# Dry Needling Alone or in Combination with Exercise Therapy versus Other Interventions for Reducing Pain and Disability in Subacromial Pain Syndrome: A Systematic Review and Meta-Analysis

**DOI:** 10.3390/ijerph191710961

**Published:** 2022-09-02

**Authors:** Gonzalo Para-García, Ana María García-Muñoz, José Francisco López-Gil, Juan Diego Ruiz-Cárdenas, Ana Isabel García-Guillén, Francisco Javier López-Román, Silvia Pérez-Piñero, María Salud Abellán-Ruiz, Fernando Cánovas, Desirée Victoria-Montesinos

**Affiliations:** 1Physiotherapy Department, Faculty of Health Sciences, Universidad Católica de Murcia, 30107 Murcia, Spain; 2Health Sciences Department, Campus de los Jerónimos, Universidad Católica San Antonio de Murcia (UCAM), Carretera de Guadalupe s/n, 30107 Murcia, Spain; 3Health and Social Research Center, Universidad de Castilla-La Mancha (UCLM), 16002 Cuenca, Spain; 4ECOFISTEM Research Group, Faculty of Health Sciences, Catholic University of Murcia, 30107 Murcia, Spain; 5Primary Care Research Group, Biomedical Research Institute of Murcia (IMIB-Arrixaca), 30120 Murcia, Spain

**Keywords:** dry needling, shoulder impingement syndrome, exercise therapy, trigger points

## Abstract

This systematic review and meta-analysis examined the effects of dry needling alone or in combination with exercise therapy for reducing pain and disability in people with subacromial pain syndrome. Systematic searches of randomized clinical trials (RCTs) were performed in five different databases. A meta-analysis was carried out with the data obtained, and the risk of bias and quality of the studies was assessed using the Cochrane ROB 2.0 and GRADE tools. Finally, five RCTs (*n* = 315) were included in the meta-analysis and qualitative analysis. Our results determine that dry needling alone or combined with exercise therapy showed improvements in pain in the short-term (5RCTs: SMD: −0.27; [−0.49, −0.05]; low-quality) and mid-term (4RCTs: SMD: −0.27; [−0.51, −0.04]; low-quality) compared to a range of interventions. However, no differences were shown for disability at short-term (3 RCTs: SMD: −0.97; [−2.04, 0.11]; very-low quality) and mid-term (3 RCTs: SMD: −0.85; [−1.74, 0.04]; very-low quality). Dry needling alone or in combination with exercise therapy may result in a slight reduction in pain in the short-term and mid-term. However, the evidence about the effect of this therapy on disability in the short- or mid-term is very uncertain compared to the range of interventions analyzed in this systematic review (Registration: INPLASY202260112).

## 1. Introduction

Shoulder pain is one of the most common musculoskeletal pains [1], with a prevalence ranging from 7 to 27% among adults younger than 70 years old [2]. Regarding the possible origin of this disorder, subacromial pain syndrome has been pointed out as one of the most frequent causes [3]. This disorder is defined as a shoulder problem, usually unilateral and non-traumatic, which is located around the acromion, and that is frequently aggravated when patients raise their arms [4]. This syndrome manifests itself in multiple forms, producing degeneration of the rotator cuff tendons and degeneration of the bursa of the subacromial space [5], however, their pathogenesis is not fully understood. Different etiological theories have been attributed to this pathology. In this sense, some studies have attributed the pathology origin to purely structural factors, changes in the arrangement of the humeral head in the glenoid cavity, or the biomechanical articular interaction between the gleno-humeral and acromio-clavicular joints; among other causes [6]. However, there are other studies that do not accept this anatomical relationship assumed in those already mentioned above and, therefore, they consider the subacromial involvement or subacromial pain syndrome as a closer approach for explaining this affection [6].

Currently, the relationship between myofascial trigger points (MTrPs) and shoulder pain is supported by different authors [7,8,9,10]. MTrPs are highly irritable points located in the taut bands of skeletal muscle, with a very sensitive response to pressure [11]. These points generate motor dysfunction and spontaneous pain, or pain on compression [12]. The scapula plays an important biomechanical role in the shoulder’s function. Therefore, the normal arthrokinematics of the gleno-humeral region could be affected due to a muscle imbalance [13]. The disability and pain caused by MTrPs can alter scapulohumeral rhythms, which may lead to subacromial pain syndrome [13]. Although the theory of the trigger points is still under discussion [14], several articles have shown a high concentration of inflammatory mediators in the subscapularis tendon and joint capsule [15,16], and a high activity in acetylcholine and nicotinic receptors [17], explaining that such conditions may produce a persistent and painful contraction.

Nevertheless, the uncertainty about the pathogenesis of this syndrome is partly reflected in the diversity and confusion of its treatment. Conservative strategies have been offered as a first choice of therapy. Indeed, numerous non-operative treatments have been proposed for improving shoulder pain syndrome-related symptoms, such as exercise therapy, manual therapy, laser therapy, ultrasound therapy, extra-corporeal shockwave, or kinesio tapping, among others [18,19]. However, the diversity in these strategies does not necessarily make the decision easier for clinicians unless a comprehensive analysis has been previously carried out for effectiveness. Exercise therapy has been recommended in a recent overview of systematic reviews as the first-line treatment to improve shoulder pain and functional disability due to its clinical effectiveness, cost-effectiveness, and other associated health benefits [18].

Currently, there is a growing tendency to use invasive approaches such as dry needling therapy either alone or combined with exercise therapy for managing subacromial pain syndrome-related symptoms. Dry needling is a minimally invasive treatment in which a small needle is inserted throughout the skin. The aim of this method is the stimulation of MTrPs, connective tissue, and muscles in order to reduce pain and functional disability. The mechanisms for producing all of these actions are not yet fully understood. However, several meta-analyses have shown that dry needling can produce a beneficial effect in reducing pain, and it may inactivate or eliminate MTrPs in shoulder pain [9]; neck pain [20]; spinal pain [21], and in several musculoskeletal disorders [22]. Different factors have been proposed as possible neurophysiological mechanisms of this condition, such as patient expectations, previous patient experience, the placebo effect, decreasing nociceptive afferences, and biochemical changes, among others [23].

A systematic review published by Blanco-Díaz et al. [24] evaluated the effectiveness of dry needling in combination with physiotherapy for the recovery of patients with subacromial pain syndrome. However, this manuscript did not perform a meta-analysis. Thus, to provide an evidence-based approach, this manuscript contributes to the scientific knowledge by adding a comprehensive analysis of the effectiveness of dry needling alone or combined with exercise therapy for reducing pain and functional disability in people with subacromial pain syndrome. Different studies, such as the meta-analysis conducted by Navarro-Santana et al. [9], noted the effect of dry needling on MTrPs in non-specific shoulder pain, producing a short-term decrease in pain. Another meta-analysis, conducted by Sánchez-Infante et al. [22], showed that dry needling decreased the pain of various musculoskeletal pathologies, including non-specific shoulder pain. Therefore, the aim of this systematic review and meta-analysis was to analyze the effects of dry needling therapy, either alone or in combination with exercise therapy, on pain and disability in people with subacromial pain syndrome. We hypothesize that dry needling alone and in combination with therapeutic physical exercise may decrease pain and disability related to subacromial syndrome.

## 2. Materials and Methods

### 2.1. Study Design

A systematic review was carried out following the Preferred Reporting Items for Systematic Review and Meta-Analyses (PRISMA) statements and following the recommendations of the Cochrane Collaboration’s Manual for Systematic Reviews of Interventions [25]. The review protocol was registered in the International Platform for Registered Protocols for Systematic Review and Meta-Analysis (INPLASY registration number: 202260112).

### 2.2. Eligibility Criteria

For the selection of the different studies considered in this publication, the following criteria was proposed:

Inclusion criteria: (a) the presence of participants with subacromial pain syndrome; (b) the presence of a group of patients receiving dry needling or dry needling in combination with exercise therapy; (c) the presence of a control group; (d) to have pain or disability as a primary outcome; (e) to be a randomized clinical trial.

The exclusion criteria included: (a) to be randomized clinical trials published as a conference abstract due to insufficient information reported and an inability for data extraction; (b) being studies focusing on electrical dry needling or studies applying dry needling interventions to both groups; (c) studies using acupuncture instead of dry needling; and (d) being studies in which shoulder pain or disability are not measured.

### 2.3. Information Sources and Search Strategy

For the preparation of this systematic review, the following computerized databases were consulted: IBECs through the Virtual Health Library platform; PUBMED through the National Center of Biotechnology Information platform; CENTRAL through the Cochrane Library platform; Web of Science Core Collection through the Web of Science platform; and Scopus. The last search was performed on 3 July 2022.

The terms “dry needling”, “shoulder impingement syndrome”, “subacromial pain syndrome”, “rotator cuff”, pain; soreness; disability, random* were combined with the Boolean operators AND/OR in the aforementioned databases. MeSH descriptors were also used in all databases when appropriate. Additionally, references from included studies were checked, looking for potential articles of interest. The search strategy is shown in Appendix B (Table A1).

### 2.4. Study Selection Process

Titles, abstracts, and conclusions were reviewed by a single author (G.P.-G.), from which irrelevant studies were removed. The remaining studies were then read by three masked, independent authors (G.P.-G., A.M.G.-M. and D.V.-M.). A.I.G.-G. was responsible for resolving discrepancies between authors. However, there were no discrepancies in the last step of the selection process. The Rayyan platform [26] was used to remove duplicate studies.

### 2.5. Data Extraction and Quality Assessment

For data extraction, a first phase was carried out by a single author (G.P.-G.) and then cross-checked point by point from the original articles by a second author (D.V.-M.) to corroborate the information extracted. For this extraction, the PICOS strategy [27] was used to obtain demographic information of the population included (i.e., age, sex, sample size, time of evolution of the injury), characteristics of the intervention (i.e., duration, sessions, frequency, exercise type, the area where dry needling was applied) for both the experimental and the control groups. Additionally, study design, country of study origin, publication date, and outcomes analyzed of each study were extracted.

The Risk of Bias 2.0 (RoB 2.0) tool proposed by the Cochrane Collaboration [28] was used in this systematic review. This tool evaluates six domains that can be rated as low risk of bias when the bias is unlikely to affect the results, as high risk of bias when the bias likely affects the results, or as unclear risk of bias when there is not enough information for rating it as high or low risk. Two independent authors (G.P.-G. and J.D.R.-C.) assessed the risk of bias of individual studies and disagreements were resolved by consensus with a third author (D.V.-M.).

Publication bias was set to 0.10 and assessed using a Funnel plot and Egger’s test [29].

Quality of evidence was estimated by using Grading of Recommendations, Assessment, Development and Evaluation (GRADE) [30]. Inconsistency, imprecision, publication bias, risk of bias, magnitude of effect, dose-response gradient, and plausible residual confounding were evaluated.

### 2.6. Synthesis Methods

To analyze the effects of dry needling alone or combined with exercise therapy compared to other types of intervention on pain and disability at short- and mid-term periods, a set of meta-analyses were performed using the DerSimonian and Laird method [31]. Forest plots were created to graphically represent the results of each study on the outcomes included [13,32,33,34,35] with the corresponding 95% CI. For this purpose, standardized mean differences (SMD) and 95% CI were calculated for each study. The pooled effect size of SMD was classified as small (0–0.20), medium (>0.20 to 0.50), or large (>0.50). Negative standardized mean difference values for the Numeric Pain Rating Scale (NPRS) and the Visual Analogue Scale (VAS) indicate a reduction in pain. Similarly, negative standardized mean difference values for Shoulder Pain and Disability Index (SPADI) and the Disabilities of the Arm, Shoulder, and Hand (DASH) indicate a benefit regarding disability. Conversely, positive standardized mean difference values for the Constant–Murley score (CMS) indicate a decrease in disability. To a better understanding, these values were multiplied by −1. Furthermore, data obtained from multiple assessments where combined as a single effect for further analysis [36]. Inconsistency among clinical trials was assessed by *I*^2^ statistic, which was classified as not important (<40%), moderate (40–60%), substantial (60–75%), and considerable (75–100%) [25]. Statistical analyses were performed using the statistical software STATA SE, version 15 (StataCorp, College Station, TX, USA). Statistical significance was set at *p* < 0.05.

## 3. Results

### 3.1. Study Identification and Selection Process

A total of 108 records were identified across selected databases and no potentially eligible articles were recovered. In total, 47 clinical trials were examined and 42 were excluded for several reasons (Figure 1). A total of five hits were fully analyzed, meting eligibility criteria.

### 3.2. General Characteristics of Included Studies

The articles used in this systematic review were randomized clinical trials (*n* = 5) published between 2012 and 2022 (the last 10 years). Most of the articles included in this systematic review were published in 2021 (*n* = 3) [13,32,35]. The included studies were conducted in Iran (*n* = 2) [13,32], Spain (*n* = 2) [33,34], and Turkey (*n* = 1) [35].

### 3.3. Sample Characteristics

This systematic review included 315 participants (58.7% women) with an age range from 35.6 to 54.3 years [13,32,33,34,35]. Patients had a minimum of three months of symptoms durations diagnosed with subacromial pain syndrome with a minimum of pain intensity of four according to the NPRS or VAS (0–10 scale). Three studies did not report symptom durations [13,32,34].

Subacromial pain syndrome was diagnosed through different methods. Imani et al. [32] used Hawkins–Kennedy tests and a Neer’s sign positive for diagnosis. In another two studies [13,35], the diagnosis was referred by the orthopedic surgeon of the hospital. Arias-Buría et al. [33] used the positive painful arc test during shoulder abduction as a diagnostic criterion; they also considered necessary for the diagnosis, at least two positive results of the following tests: drop arm test, lift-off test, Hawkins–Kennedy test, or empty can test. Finally, Pérez-Palomares et al. [34] used ultrasound and magnetic resonance imaging to confirm the diagnosis.

### 3.4. Intervention Characteristics

The experimental protocol was based on a combination of dry needling plus exercise therapy [32,33,34] or dry needling alone [13,35] whereas the control protocol was based on a variety of interventions such as exercise therapy [33], stretching [13], massage [35] or a combination of all of them [34] plus hot packs, and analgesic electrotherapy [32].

Dry needling intervention was applied to trigger points up to local twitch muscle response was obtained using the fast-in and fast-out technique described by Hong [37] in all included studies. The treatment area was heterogeneous, with supraspinatus and infraspinatus muscles the most commonly treated. Two or three sessions of dry needling therapy were used over a 3–4-week period, with the exception of one study that applied three sessions in one week [13]. Session duration was not mentioned in most of the included studies. Some studies stated that session duration finished when local twitch muscle response was obtained, or else the needle was maintained for 10 min in a static position after the muscle response.

The exercise intervention consisted of semi-supervised and non-supervised strengthening exercises twice daily or twice weekly for five weeks ranging from 20 to 30 min duration [33,34]. However, most studies did not mention the FITT principle of training, i.e., frequency, intensity, time, and type of exercise. Arias-Buría et al. [33] performed three series of twelve repetitions of three shoulder exercises (not mentioned) at low intensity (unspecified), whereas Pérez-Palomares et al. [34] performed isometric, proprioceptive, and functional exercises applied in an individualized manner, based on the patient’s condition (Table 1).

### 3.5. Measures

Three of the included studies analyzed pain and disability as a primary outcome measure [32,33,34] and two studies only analyzed pain measures [13,35].

Three studies used the VAS ranging from 0 to 10 cm [13,34,35], whereas two studies used the NPRS ranging from 0 to 10 points [32,33] for pain measures. In both scales, a greater score obtained represents greater pain.

Regarding disability measures, a study used the SPADI [32], whereas the remaining articles used the DASH questionnaire [33] and the CMS [34]. All disability questionnaires ranged from 0 to 100 percent. Higher DASH and SPADI scores represented greater disability. Lower CMS scores represented greater disability.

### 3.6. Risk of Bias

High risk was considered when analyzed studies showed a high risk of bias in at least two domains [13,32,33,34] (Figure 2). Uncertain risk was considered for those studies that did not provide sufficient information on allocation masking (Figure 3) [13,32,35].

4 out of 5 studies showed incomplete results and therefore were rated as low risk of bias, whereas, due to insufficient protocol information in each study, selective reporting bias could not be adequately assessed in most of the included studies.

One study showed high bias in other sources because some authors had a conflict of interest [13]. Additional information can be found in the Appendix A.

### 3.7. Intervention Results

The meta-analysis found that the dry needling group alone or combined with exercise therapy improved pain intensity in the short-term and suggested that dry needling had a medium effect (SMD: −0.27; 95% CI: −0.49 to −0.05; *I*^2^ = 0.00%; *p* < 0.02; low-quality evidence) compared to other types of interventions, such as exercise therapy [33], stretching [13], massage [35], or a combination of all of them [34] plus hot packs and analgesic electrotherapy [32] with no heterogeneity (Figure 4).

There was a significant decrease in pain intensity at the mid-term. The meta-analysis suggest that dry needling had a medium effect (SMD: −0.27; 95% CI: −0.51 to −0.04; *I*^2^ = 0.00%; *p* < 0.02; low quality evidence) (Figure 5).

Regarding disability, the meta-analysis found no significant differences between the dry needling group alone or combined with exercise therapy compared to other types of interventions, such as exercise [33] or a combination of exercise, massage, passive mobilization [34] plus hot packs and analgesic electrotherapy [32] in the short-term and suggest that dry needling had a large effect (SMD: −0.97; 95% CI: −2.04 to 0.11; *I*^2^ = 93.11%; *p* = 0.08; very-low quality evidence) (Figure 6).

However, no significant differences between groups were found on disability at the mid-term. The meta-analysis suggest that dry needling had a large effect (SMD: −0.85; 95% CI: −1.74 to 0.04; *I*^2^ = 89.44%; *p* = 0.06; very-low quality evidence) (Figure 7).

Studies were evenly distributed through the funnel plot for the outcome variable (Appendix A). However, for the outcome disability, the funnel plot was asymmetric (Appendix A). Funnel plots can be found in the Appendix A.

### 3.8. Quality of Evidence

The quality of evidence for the effectiveness of dry needling alone or in combination with exercise therapy compared to other types of treatments on pain at short- and mid-term in subacromial pain syndrome was rated as low quality, whereas the quality of evidence on disability in the short- and mid-term was rated as very-low quality. Reasons for rating down the quality of evidence were due mainly to a lack of blinding of participants, personnel, and outcome assessors and imprecision.

## 4. Discussion

The objective of this systematic review was to analyze the effects of dry needling alone or in combination with exercise therapy on pain and disability in people with subacromial pain syndrome. Overall, the results of this study showed that dry needling alone or combined with exercise therapy may provide small benefits on pain compared to other forms of treatments such as exercise therapy, stretching, massage, or a combination of all of them plus hot packs and analgesic electrotherapy in the short-term (<5 weeks) and mid-term. In addition, it has been observed that this treatment produces a significant decrease in pain in the mid-term (from one to 12 months). Our pooled results showed no difference between the experimental and the control group regarding disability measures neither in the short- and mid-term. Although better results were obtained with dry needling, these differences were not statistically significant. The low number of studies included in the present systematic review with meta-analysis could explain the lack of statistical significance.

Most of the included studies were rated at high risk of bias, mainly due to a lack of blinding of the participants, personnel, and outcome assessors. Most studies reported that the investigator assessing outcomes was masked; however, pain and disability were measured by self-reports from participants who could not be masked. Blinding participants in clinical trials with self-report measures is crucial to not overestimate the intervention effect, otherwise, results on pain and disability may be overestimated by up to 25% [38].

In a comprehensive meta-analysis, Braithwaite et al. [39] focused on the influence of inadequate blinding on pain in dry needling trials and reported that the intervention effect was in favor to dry needling groups compared to sham groups when inadequately blinded group comparisons were performed, but no differences were reported during adequately blinded group comparisons. Lack of adequate blinding leads to an undistinguished effect between dry needling and the effect provided by bias (e.g., placebo effect). Therefore, despite this intervention potentially providing a small benefit on pain in the short-term, the results should be interpreted with caution. In fact, the quality of evidence was rated as low to very-low, thus, the true effect may be substantially different from the estimate of the effect [40].

Our results were partially similar to those recently reported by a systematic review and meta-analysis examining the effects of dry needling on pain and disability in nontraumatic shoulder pain [9]. They reported small improvements in pain in the short term in favor of the dry needling groups. However, no differences were reported during the follow-up period. Contrary to our findings, they reported an overall large effect (SMD = −1.14; 95%CI: −1.81 to −0.47) in favor to the dry needling groups for reducing disability in these patients. Differences between studies could be (at least) partially explained because in their meta-analyses, data from multiple-time points were not combined into a single effect, therefore, this approach assigns more weight to studies with multiple-time points than those with a single-time point. Additionally, this approach leads to an improper estimation of the precision of the summary effect since it treats different time points as independent of each other, when in fact, they come from the same set of individuals [9]. This was evident when two studies with positive findings for the dry needling group were treated as five independent studies [9], so the precision and magnitude of the overall effect could be overestimated.

Similar to our results and those obtained by Navarro-Santana et al., Liu et al. [10] found a beneficial effect of dry needling on MTrPs in the short term [9]. However, they did not achieve significant results at follow-up after two months. One possible reason for this discrepancy could lie in the population analyzed that Liu et al. [10] included, in addition to the use of dry needling and acupuncture, indistinctly. Conversely, Hall et al. [41] conducted a meta-analysis that included subjects with shoulder pain and, contrary to our findings, they did not obtain strong evidence about the decrease of shoulder pain. Their result could be explained due to the different etiologies of the selected patients suffering the disorder.

Multiple mechanisms of action have been attributed to dry needling in order to explain its role in the pain decrease. Dry needling could increase the blood flow in MTrPs, thus producing a decrease in sensitizing substances such as bradykinin and calcitonin [17,42]. It could also reduce synaptic [43] and pH alterations, providing a normal level of enzymes such as acetylcholinesterase in the place of action, and thus, decreasing the dysfunction of the motor plate of the MTrPs [17,43]. Another theory states that dry needling can produce antinociception through the release of endogenous opioid peptides. This effect might improve the hyperalgesia condition suffered by the patients [44]. There are diverse factors that can influence the mechanism of action of dry needling, such as the place of needle application, the time of the treatment, or the use of adjuvant methods, among others. All of them could, at least partially, explain the wide diversity of results obtained after the dry needling strategy.

A recent summary of systematic reviews was conducted to examine the efficacy of conservative physiotherapy intervention in subacromial syndrome pain [18]. Conclusions state that therapeutic exercises should be considered one of the best interventions in the treatment of this syndrome. However, further studies should establish the most appropriate exercise to reduce associated symptoms. This issue is mainly due to insufficient information provided by the included studies. In our systematic review, while the studies were focused on providing information about the dry needling protocol, most of the included studies did not provide information about the frequency, intensity, time, and type of exercise (*FITT* principle) or whether it requires supervision or individualization, so precise information is required to fully understand the intervention effect and how to replicate it. To avoid this issue, further studies are strongly recommended following the Consensus on Exercise Reporting Template (CERT) [45].

Due to its clinical effectiveness, cost-effectiveness, and other associated health benefits, exercise therapy has been recommended as the first-line treatment to improve shoulder pain and disability [18]. This systematic review is not aimed at evaluating cost-effectiveness, however, one study reported that dry needling plus exercise therapy might reduce the costs associated with the absenteeism paid labor compared to exercise therapy alone [46]. However, our meta-analysis showed small improvements in pain in the short and mid-term. Nonetheless, evidence is scarce and further studies should be performed for a better understanding of dry needling, plus exercise therapy’s cost-effectiveness in patients with subacromial pain syndrome.

This review presents some limitations. Several protocols were found through unpublished systematic searches, increasing the risk of publication bias. However, these unpublished studies likely do not exaggerate the net benefit of the intervention; this is the reason why we did not downgrade the quality of the evidence [47]. Additionally, since most studies did not blind the participants and were focused on self-report measures (performance and detection bias), a sensitive analysis to elucidate the influence of adequate blinding was not possible. The different strategies used for the subacromial pain syndrome diagnosis found in the consulted studies result in another limitation for the development of our meta-analysis. This fact could introduce some risk of bias. Therefore, the results of this systematic review are affected by these biases, so the self-reported improvements in pain in the short and mid-term could be overestimated. Conversely, to our knowledge, this is the first systematic review with metanalysis assessing the reduction of shoulder pain and disability by dry needling alone or in combination with therapeutic exercise.

## 5. Conclusions

Dry needling alone or in combination with exercise therapy may result in a slight reduction in pain in the short-term and mid-term compared to the range of interventions analyzed in this systematic review with meta-analysis. However, there is not enough evidence of the effect of dry needling alone or combined with exercise therapy on disability in the short- or mid-term.

## Figures and Tables

**Figure 1 ijerph-19-10961-f001:**
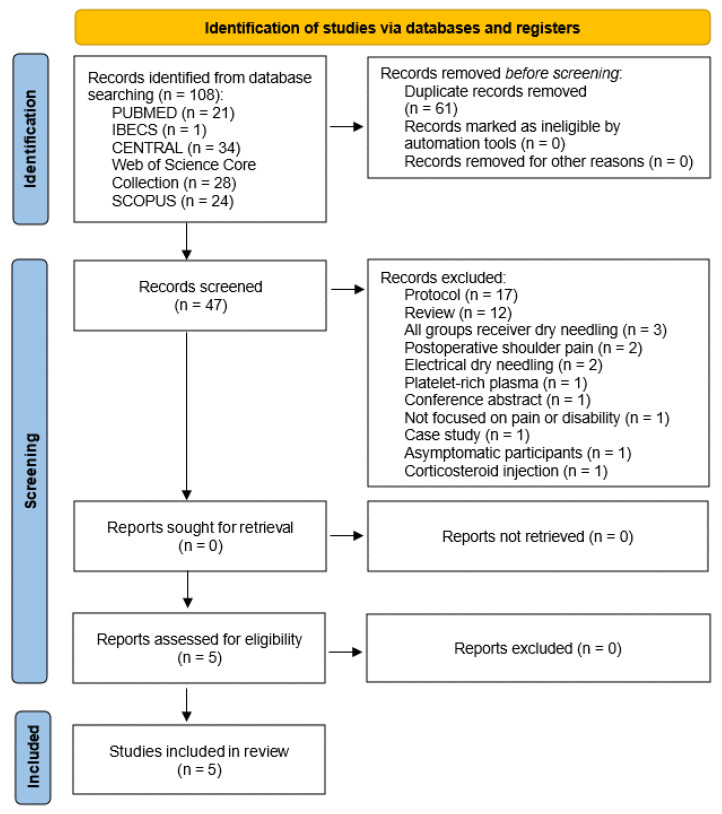
Flowchart of the study selection process.

**Figure 2 ijerph-19-10961-f002:**
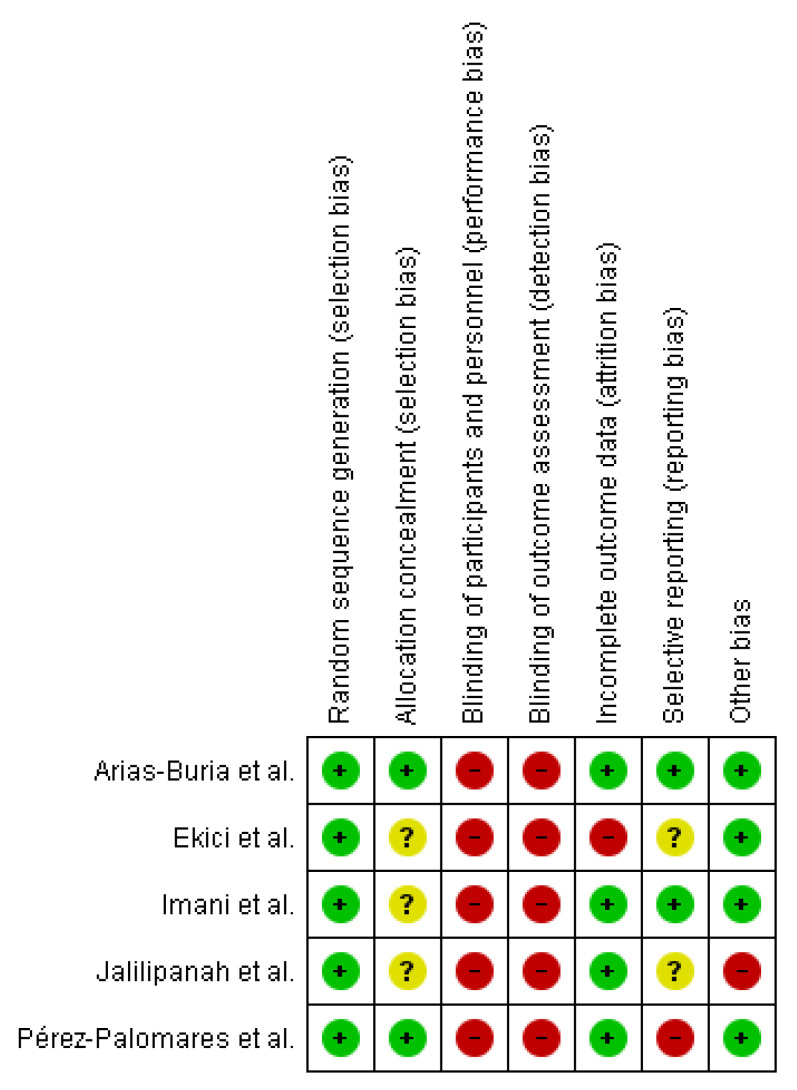
Risk of bias summary: a review of authors’ judgments about each risk of bias item for each included study [13,32,33,34,35]. Red (−) = high risk of bias, yellow (?) = unclear risk of bias, green (+) = low risk of bias.

**Figure 3 ijerph-19-10961-f003:**
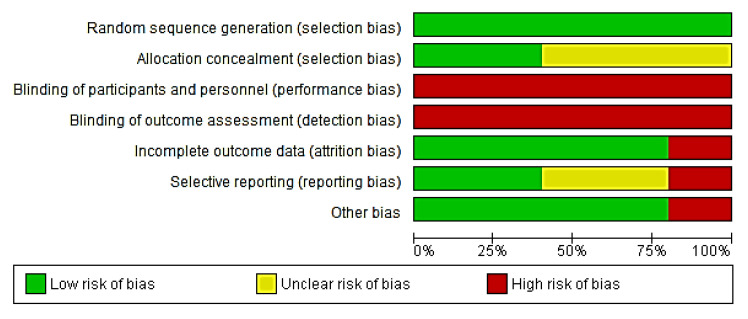
Risk of bias graph: a review of authors’ judgments about each risk of bias item presented as percentages across all included studies [13,32,33,34,35].

**Figure 4 ijerph-19-10961-f004:**
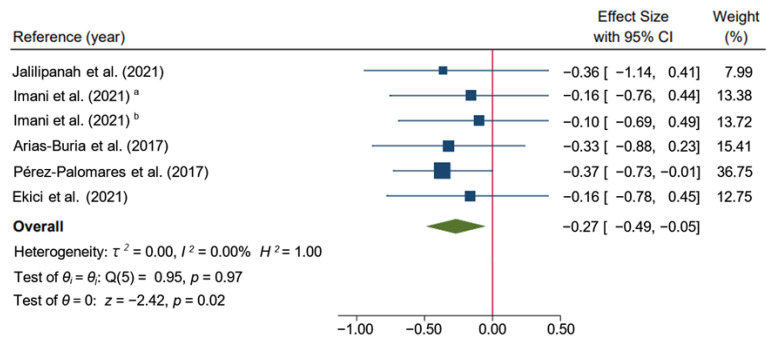
Forest plot of comparison in the short-term: Dry Needling group alone or combined with exercise therapy versus other types of intervention (exercise therapy, stretching, massage, or a combination of all of them plus hot packs and analgesic electrotherapy), outcome: Pain. ^a^: Hong’s dry needling technique (fast-in fast-out intervention); ^b^: Deep dry needling (static needling) [13,32,33,34,35].

**Figure 5 ijerph-19-10961-f005:**
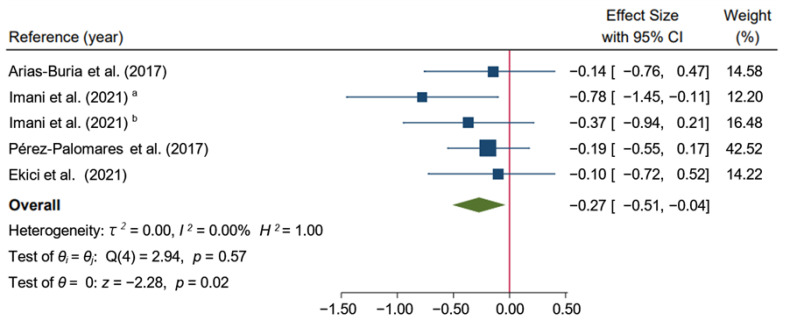
Forest plot of comparison at the mid-term: Dry Needling group alone or combined with exercise therapy versus other types of intervention (exercise therapy, stretching, massage, or a combination of all of them plus hot packs and analgesic electrotherapy), outcome: Pain. ^a^: Hong’s dry needling technique (fast-in fast-out intervention); ^b^: Deep dry needling (static needling) [32,33,34,35].

**Figure 6 ijerph-19-10961-f006:**
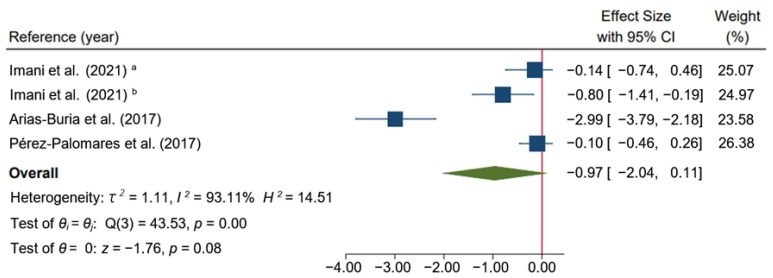
Forest plot of comparison in the short-term: Dry Needling group alone or combined with exercise therapy versus other types of intervention (exercise, or a combination of exercise, massage, passive mobilization plus hot packs and analgesic electrotherapy), outcome: Disability. ^a^: Hong dry needling technique (fast-in fast-out intervention); ^b^: Deep dry needling (static needling) [32,34,35].

**Figure 7 ijerph-19-10961-f007:**
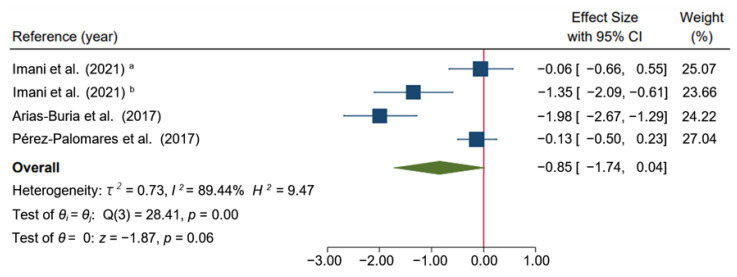
Forest plot of comparison at the mid-term: Dry Needling group alone or combined with exercise therapy versus other types of intervention (exercise, or a combination of exercise, massage, passive mobilization plus hot packs and analgesic electrotherapy), outcome: Disability. ^a^: Hong dry needling technique (fast-in fast-out intervention); ^b^: Deep dry needling (static needling) [32,34,35].

**Table 1 ijerph-19-10961-t001:** Characteristics of the included studies.

	Intervention								Comparison			
	Dry Needling				Exercise Therapy				Control Group			
Studies	Subjects	Age Mean	Shoulder Pain Duration	Protocol	Area	Duration	Protocol	Duration	Subjects	Age Mean	Shoulder Pain Duration	Protocol
Arias-Buría et al. [33]	EG: *n* = 25 (28% W)	49.0 (5.0)	≥4 months	TrP-Dry needling	Anterior and middle deltoid, supraspinatus, infraspinatus, teres minor and major, and subscapularis	Twice daily (five weeks)	Three sets of 12 repetitions. Concentric and eccentric exercises	20 to 25 min	CG: *n* = 25 (24% W)	48.0 (6.0)	Four months	Eccentric and concentric exercises
Imani et al. [32]	EG: *n* = 21 (71.4% W)	44.4 (9.0)	Not reported	Hong’s dry needling technique	Infraspinatus muscle	Third, fifth, and seventh session of treatment	Not included	Not included	CG: *n* = 22 (59.1% W)	41.9 (10.1)	Not reported	Analgesic electrotherapy
Ekici et al. [35]	EG: *n* = 29 (61.9% W)	52.0 (9.0)	≥3 months	Dry needling up to local twitch response	Infraspinatus, supraspinatus, subscapularis, the upper part of the trapezius and levator scapula muscles	Six treatments over a four-week period	Not included	Not included	CG: *n* = 31 (90% W)	50.9 (7.9)	≥3 months	Deep friction massage to analgesia
Jalilipanah et al. [13]	EG: *n* = 13 (77% W)	35.6 (8.7)	Not reported	Dry needling up to local twitch response	Infraspinatus muscle	Three sessions in a one-week	Not included	Not included	CG: *n* = 13 (46% W)	36.6 (7.9)	Not reported	Post-isometric relaxation
Pérez-Palomares et al. [34]	EG: *n* = 57 (70% W)	52.7 (11.8)	Not reported	Dry needling up to local twitch response and physical therapy	Supraspinatus, infraspinatus, sub-scapularis (medial, lateral superior, and lateral inferior), teres minor, and deltoid (anterior, medial, and posterior) muscles	First, fourth, and seventh session (three sessions)	Exercise and passive mobilization	30 min	CG: *n* = 63 (56% W)	54.3 (11.5)	Not reported	Exercise, massage, and passive mobilization

Data are given as mean (standard deviation) or number (percentage). Experimental Group (EG); Control Group (CG); Women (W); Trigger points (TrP).

## Data Availability

Not applicable.

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
