# Peer review of "Dry Needling Alone or in Combination with Exercise Therapy versus Other Interventions for Reducing Pain and Disability in Subacromial Pain Syndrome: A Systematic Review and Meta-Analysis"

_ijerph, 2022, doi:10.3390/ijerph191710961_

Round 1

Reviewer 1 Report

The topic is of interest, but the paper is out of date about current data on dry needling for the shoulder. 

The title is confusing since it seems that authors compared dry needling alone versus dry needling combined with exercise, when the reality is that they compared dry needling (alone or combined) versus other interventions. Please edit the title being more precise. 

The term subacromial impingement is out of date. Authors should use subacromial pain syndrome and updated current theories on this syndrome. Several papers are published on this topic. In addition, the introduction does not include previous MA on this topic. Two MA published in PT Journal about dry needling, one just on shoulder pain and one on MSK conditions have been published. Similarly, authors should include previous SR on the topic such as one published in physiotherapy journal. In conclusion, the introduction section should be completely rewritten and including a clear justification for this MA. What is the new information provided by this MA in relation to current data published and not discussed in the paper? 

The selection criteria should follow PICO statement. The methods section does not follow PRISMA. Authors should define their follow-up periods: short, mid and long-term. Intermediate is not a common term for defining follow-up period. 

The results section again jumps from one to another topic and does not adhere to PRISMA recommendations. Please reorder. 

Figures: which different exist between Hong dry needling and deep dry needling?  In some figures the study Imani et al is repeated without explanation and this is a bias in the results. 

Reviewer 2 Report

Thank you for letting me review this interesting manuscript about the effects of dry needling in subacromial impingement. In general, I think is a well-written paper, and the methods and results are quite clear. 

I have few suggestions and comments that should be adressed before further consideration: 

- Line 56-57. This sentences does not fit with the other paragraphs. Please, delete it or include it in the previous paragraph. 

- Line 59-60. A trigger point hyothesis should be included. Otherwise, the use of invasive techniques is not justified. 

- Inclusion criteria. Please describe the PICOs section according to the study

- exclusion criteria. A more detailed explanation of ht exclusion criteria is needed. 

- Line 112. Was the Kappa coefficient calculated?

- Line 113-114. The authors said tht used 3 independent authors to review the RCTs. However, then is written that a third independent author solved the incongruences. Should be the fourth author? Please clarify this question. 

- Sample characteristics. How was the impigement syndrome diagnosed in each study?

- Discussion. the hyopothesis about the mechanism of action of dry needling in subacromial impingemetn should be discussed. 

Round 2

Reviewer 1 Report

Authors have properly answered to all comments, well done

Author Response

Thank you very much for your comments.

Reviewer 2 Report

Thank you for taking the time to implement all the sugestions. I have few minor comments:

Line 38-40. Both sentences have the same meaning nad the word "among" is repeated. Please revise. 

Line 56- How is the relationship between the MTrP and shoulder impingement. It is not well introduced. 

Line 200. The different clinical tests used for the subacromial impingement diagnosis should be consider as a limitation. Please add a sentenece in the limitations paragraph. 
